

# An adapted hourly Himawari-8 fire product for China:principle, methodology and verification.

Jie Chen[1],[★], Qiancheng Lv[2],[★], Shuang Wu[3], Yelu Zeng[4], Manchun Li[5], Ziyue Chen[2] , Enze Zhou[6], Wei Zheng[1], Cheng Liu[2], Xiao Chen[2], Jing Yang[2], and Bingbo Gao[4]

[1]Key Laboratory of Radiometric Calibration and Validation for Environmental Satellites National Satellite Meteorological Center (National Center forSpace Weather), China Meteorological Administration and Innovation Center for FengYun Meteorological Satellite (FYSIC), Beijing 100081, China
[2]College of Global Change and Earth System Science, Beijing Normal University, Beijing 100091, China
[3]Heilongjiang Eco-Meteorology Center, Harbin,Heilongjaing 150030, China
[4]College of Land Science and Technology, China Agricultural University, Beijing 100083, China
[5]School of Geography and Ocean Science, Nanjing University, Nanjing 210023, China
[6]Electric Power Research Institute Gongdong Power Grid, Guangzhou, Guangdong 510000, China
★ These authors contributed equally to this work.

*Correspondence to*: Ziyue Chen (zychen@bnu.edu.cn) and Bingbo Gao (gaobingbo@cau.edu.cn)

**Abstract.** Wild fires exerts strong influences on the environment, ecology, economy and public security. However, the existing hourly JMA-Himawari fire product presents large uncertainties and not suitable for reliable real-time fire monitoring in China. To fill this gap, we proposed an adaptive hourly NSMC-Himawari fire product for China based on the original Himawari source by employing a dynamical threshold for fire-extraction and a database of ground thermal sources. According to the visually-extracted reference and consistence check, we found NSMC-Himawari fire product effectively removed a majority of false fire alarms included in the JMA-Himawari fire product. Based on a rare field-collected ground reference dataset, we evaluated the reliability of JMA-Himawari and NSMC-Himawari fire product across China. The overall accuracy of JMA- Himawari fire product was 54 % and 59 % (not considering the omission errors) respectively. As a comparison, by identifying more real fire pixels and avoiding a majority of false fire alarms, the overall accuracy of NSMC-Himawari fire product was 80 % and 84 % (not considering the omission errors) respectively, making it an ideal source for improved real-time fire monitoring across China. This research also provides useful reference for employing local dataset of underlying surfaces and thermal sources to enhance the accuracy of global fire products in specific regions. NSMC-Himawari fire product can be downloaded at http://figshare.com (last access: 14 November 2022) with the identifier DOI: https://doi.org/10.6084/m9.figshare.21550248 (Chen et al., 2022).

## 1 Introduction

With a vast coverage, wild fires across China have a strong negative impact on crops (Song et al., 2022), forests (Ying et al., 2018; Ni1 et al., 2012; Zackrisson, 1977), grasslands (Balch et al., 2013), biodiversity (Driscoll et al., 2010; Andersen, et al., 2005), wildlife (Tiedemann et al., 2000), local and regional climate (Liu et al., 2018; Marlier et al., 2015) and public health



(Huff et al., 2015; Johnston et al., 2012). Recently, increasing occurrence of annual forest fires in China has caused a large number of causality (Wu et al., 2014; Liu et al., 2012). The cross-country grassland fires, especially grasslands at the China-Mongolia boundary, are exerting an increasing serious threat to the sustainable development of grasslands (Na et al., 2018), the wealth (Abram et al., 2021; Cascio, 2018; Reid et al., 2016) and lives (Liu et al., 2015) of local residents. In addition to natural fires, anthropogenic fires, majorly the fires caused by crop-residue burning, which has been considered as one significant driver of airborne pollution (Li et al., 2015; Huang et al., 2012; Angassa and Oba, 2008) and strictly controlled across China, are given growing emphasis in recent years.

Given the significant role of wild fires in the national ecology, environment and economy, as well as people's daily, both the central and local governments are making substantial efforts on the preventing and controlling natural and human-induced fires (Panjaitan et al., 2019). Amongst a series of measures, the timely and precise monitoring is key for the effective identification and suppression of wild fires (Earl and Simmonds, 2018; Andela et al., 2017; Zhang et al., 2014). In the past, regular forest, farmland and grassland inspection is the major approach for preventing wild fires (Röder et al., 2008). However, this approach is highly resource-consuming and of extremely low efficiency (Giglio et al., 2006). Furthermore, for those distant regions with terrible natural conditions and scarce residents, it is difficult for people to conduct manual inspection. In that case, wild fires occur in these areas can hardly be detected within a short period. To address this issue, some alternative solutions have been employed. For instance, a large number of on-site monitoring sensors can be installed randomly in forests or grasslands to real-time capture the smoke and heat and thus quickly detect active fires (Li et al., 2019). However, massive resources are required for installing and maintaining these sensors. Meanwhile, there is still a majority of areas that cannot be covered by these sensors and thus the general efficiency and accuracy of fire monitoring remains low (Ichoku et al., 2012; Freeborn et al., 2011).

Thanks to its easy access, low cost, continuous time series and most importantly, large-scale and spatially continuous coverage, remote sensing has been increasingly employed for fire monitoring (Xu et al., 2017; Schmit et al., 2017). Since 1970s, National Oceanic and Atmospheric Administration (NOAA) series satellites (www.noaa.gov) and NOAA fire products, with a spatial resolution of 1.1 km and a daily temporal resolution, have been employed for global fire monitoring (Fuller and Fulk, 2018). Afterward, many regional or global fire products has been proposed in recent years. Attributed to its free access, long time series, and reliable accuracy (Giglio et al., 2018), Moderate Resolution Imaging Spectroradiometer (MODIS) fire product, with a spatial resolution of 1km and a temporal resolution of 12 hours, have become one of the most widely employed fire products since 2000 (Justice et al., 2002). Visible Infrared Imaging Radiometer Suite (VIIRS) fire products, with a temporal resolution of 12 hours and an improved spatial resolution of 375 m, has been available since 2011 (Schroeder et al., 2014). Despite a higher spatial resolution, the accuracy of VIIRS fire products is generally lower than that of MODIS fire products (Sharma et al., 2017). In recent years, China has launched a series of Fengyun series satellites, based on which a diversity of meteorological and fire products can be produced. We recently publicly released a Fengyun-3D global fire product (http://satellite.nsmc.org.cn/portalsite/default.aspx, Chen et al., 2022), with a daily resolution and 1km spatial resolution. Fengyun-3D fire product has a good global consistence with MODIS fire product and presented a

significantly improved accuracy in China (Chen et al., 2022), making it an ideal continuity of MODIS fire products and a better choice for relevant study in China.

Despite the rapid development of daily-level and moderate-resolution global fire products, they are not suitable for quick
identification of wild fires and understanding their environmental and ecological consequences (Yang et al., 2019). To fill this gap, specific satellites have been launched recently to produce fire products with extremely high temporal resolution. GEOS-16 Advanced Baseline Imager (ABI) fire products, with a temporal resolution of five minutes and a spatial resolution of 2 km, have been available for regional fire monitoring in Southeastern Conterminous United States (CONUS) since 2017 (Hall et al., 2019). Himawari-8 products from Japan Meteorological Agency (JMA) , with a spatial resolution of 2 km and
temporal resolution of 10 minutes, have been employed to monitor meteorology and wild fires in Asia and Australia since 2015 (Xu and Zhong, 2017). Given its moderate spatial resolution, extremely high temporal resolution and its fixed observation region, Himawari-8 is quite suitable for real-time monitoring wild fires in China. Furthermore, compared with MODIS or Fengyun-3D fire products, which fail to capture the occurrence of in the night, Himawari-8 is advantageous of continuously monitoring wild-fires after sunset, which is highly suitable for replacing manual inspection to monitor wild
fires in the night. However, existing Himawari-8 fire product presented a poor consistence with MODIS data (Jang et al., 2019) and the accuracy of Himawari-8 fire products in China remained unknown Without reliable accuracy report, Himawari-8 fire products, despite its good potential, have been limitedly employed (Na et al., 2018).

To provide a more reliable fire product for active fire monitoring in China, based on the raw Himawari-8 data, we proposed a new hourly fire product for China (National Satellite Meteorological Center Himawari-8 fire product for China, NSMC-
Himawari fire product for China) by employing adapted fire detection algorithms and local parameters concerning underlying surface and meteorological conditions, which were not available and implemented by JMA. This paper introduces the characteristics and fire detection algorithms of the new fire product (recently downloadable at http://figshare.com). Meanwhile, based on the ground-truth data, which is rarely collected and employed for verifying fire detection in China, we compared the accuracy of JMA-Himawari and NSMC-Himawari fire product. Thanks to its good
spatiotemporal resolution and significantly improved accuracy, NSMC-Himawari fire product has the potential to largely prompt the fire prevention and control and boost relevant ecological and environmental research in China.

## 2 Overview of Himawari-8 fire products

### 2.1 Instrument

Himawari−8 geostationary meteorological satellite was launched by the Japan Meteorological Agency (JMA) at the
Tanegashima Space Center in October 2014, and officially began operation in July 2015 (Bessho et al., 2016). The satellite was designed to continuously support monitoring storm clouds, typhoon movements, volcanoes with continuous eruptions and other disaster prevention areas for more than 15 years. The sub satellite point is located over the equator at 140.7 ° E, with an operating altitude of approximately 35800 km. As a new generation of geostationary meteorological satellite,



Himawari−8 with an improved Advanced Himawari Imager (AHI) has a great improvement in observation frequency and
spatial resolution compared with the previous satellites. AHI includes a total of 16 bands from visible light to far infrared (as
shown in Table1), which can access the 500 m high spatial resolution in red band and support the observation time interval
of 10 min. Himawari−8/AHI with six observation spots can monitor four regions in total, including the full disk (60° N-60°
S, 80° E-160° W), Japan (scope of two Japanese regions), the tropical cyclone sensitive area (target area) and the corner of
Australia (landmark area).

**Table 1. Observation bands of Himawari−8.**

| Band | Wavelength (μm) | Waveband | Resolution (km) |
|---|---|---|---|
| 1 | 0.470 63 | Visible light | 1 |
| 2 | 0.510 00 | Visible light | 1 |
| 3 | 0.639 14 | Visible light | 0.5 |
| 4 | 0.856 70 | Near infrared | 1 |
| 5 | 1.610 10 | Near infrared | 2 |
| 6 | 2.256 80 | Near infrared | 2 |
| 7 | 3.885 30 | Thermal infrared | 2 |
| 8 | 6.242 90 | Thermal infrared | 2 |
| 9 | 6.941 00 | Thermal infrared | 2 |
| 10 | 7.346 70 | Thermal infrared | 2 |
| 11 | 8.592 60 | Thermal infrared | 2 |
| 12 | 9.637 20 | Thermal infrared | 2 |
| 13 | 10.407 30 | Thermal infrared | 2 |
| 14 | 11.239 50 | Thermal infrared | 2 |
| 15 | 12.380 60 | Thermal infrared | 2 |
| 16 | 13.280 70 | Thermal infrared | 2 |

## 2.2 Product overview

Thanks to Himawari−8/AHI's multi-band setting and the substantial improvement of high-frequency characteristics of
observation, JMA has generated a wildfire data at a spatial resolution of nominal 0.02° (2 km x 2 km at nadir) and hourly



temporal resolution (Wickramasinghe at al., 2016). This product has been widely used in wildfire monitoring and analysis in
Asia and Australia (Y. Kurihara et al.,2020), with surprising results. This product provides the location and the fire
radioactive power (FRP) of hot spots retrieved from the IR imageries carried on Himawari-8, using the retrieval algorithm
developed by Japan space agency Earth Observation Research Center (JAXA/EORC). Based on the notable difference of the
brightness temperature between fire spots and background, this JMA fire detection algorithm can reliably monitor hot spots
for the full disk (60° N-60° S, 80° E-160° W). And the reliability level of fire detection is given according to detection
situations such as sun glint, solar angle, spatial variability of the brightness temperature, and so on. JMA- Himawari fire
product has extremely high temporal resolution and good accuracy in Japan (Wickramasinghe at al., 2016). However, given
the large variations in underlying surfaces and potential limitations of the fire detection algorithm, the reliability of JMA-
Himawari fire product across the entire monitoring region remains unclear (Liu at al., 2018). Specifically for China, in these
years, increasing emphasis has been placed on the timely and reliably monitoring of wildfires, especially crop-residue
burning, and the use of satellite data sources has been widely employed. However, the consistence between multiple
mainstream fire products in China was relatively low (Liu at al., 2018), which caused large uncertainty and difficulty in
providing reliable reference for fire monitoring. With the highest temporal resolution, Himawari-based fire products have a
great potential to support the real-time fire identification in China. Nevertheless, in practical implementation, JMA-
Himawari fire product seems to have a poor consistence with commonly employed MODIS fire product in China.
Meanwhile, due to the lack of ground-truth reference, the accuracy of JMA-Himawari fire product has yet been verified and
the wide implementation of JMA-Himawari fire product in China's fire monitoring has been limited.

To fully explore the advantage of Himawari−8/AHI for fire detection and overcome the potential limitation of JMA-
Himawari fire product, we presented a new hourly fire product from 2019 to 2021, NSMC-Himawari fire product for China
with an hourly temporal resolution and a spatial resolution of 2 km. By specifically considering the underlying surface
information in China and adjusting fire detection algorithms, NSMC-Himawari fire product aims to present an improved
reliance for timely fire identification in China, which can provide fire location, burning area, intensity, district name and
distribution of underlying surfaces. As briefly demonstrated in Fig. 1, there is a large difference between the two products,
characterized with many more fire pixels identified by JMA-Himawari fire product.

The NSMC-Himawari fire product has been produced to have a better accuracy than JMA-Himawari fire product through the
following strategies. Firstly, by considering the underlying surface information, conventional thermal anomalies caused by
industrial production and photovoltaic power plants, which may be easily recognized as fires, can be effectively avoided.
Second, we dynamically adjusted the function of fire pixels identification threshold. As a comparison, commonly employed
fire-extraction algorithms used a fixed threshold when calculating the brightness temperature difference between the target
pixel and the surrounding pixels, which may cause increased uncertainty for non-uniform underlying surface, where the
difference of brightness temperature between the target pixel and surrounding pixels varied notably (Chen et al., 2022).
Thirdly, we adopted a more efficient approach to calculate the background brightness temperature. For traditional algorithms,
due to the coarse resolution of underlying surface, the influence of vegetation type, vegetation coverage and cloud coverage

cannot be fully considered. By including more reliable information of the underlying surface in China, our algorithm can better calculate the background brightness temperature of the target area.

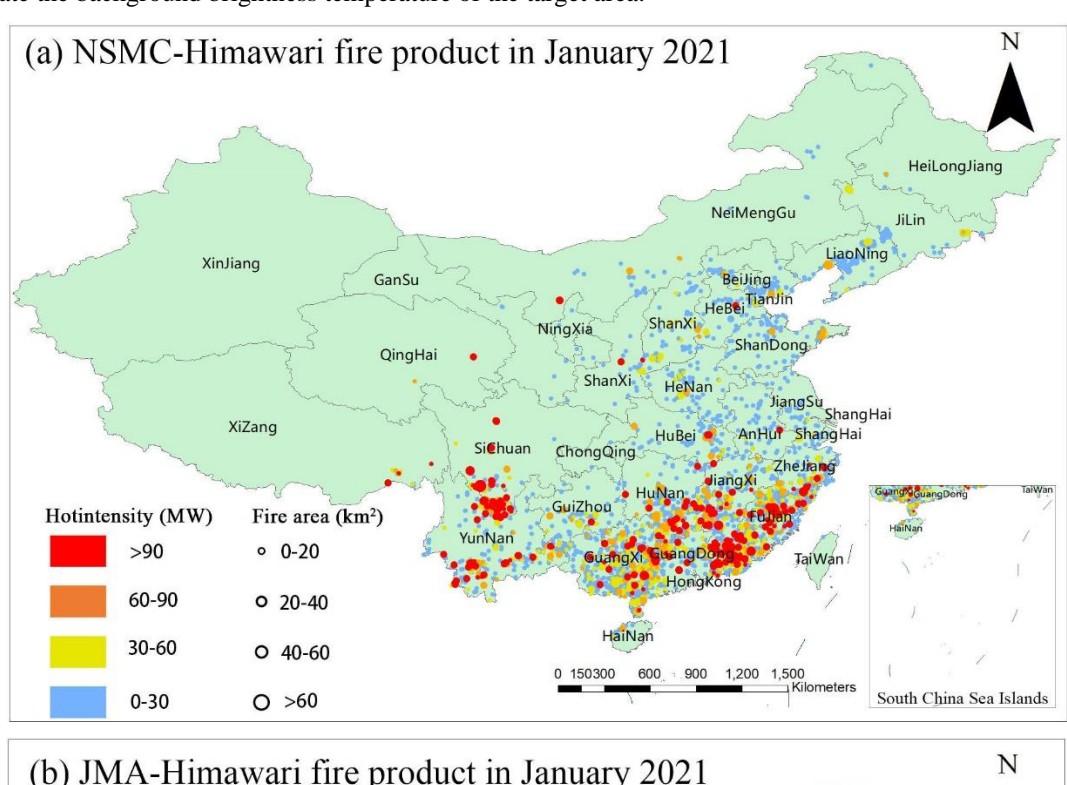

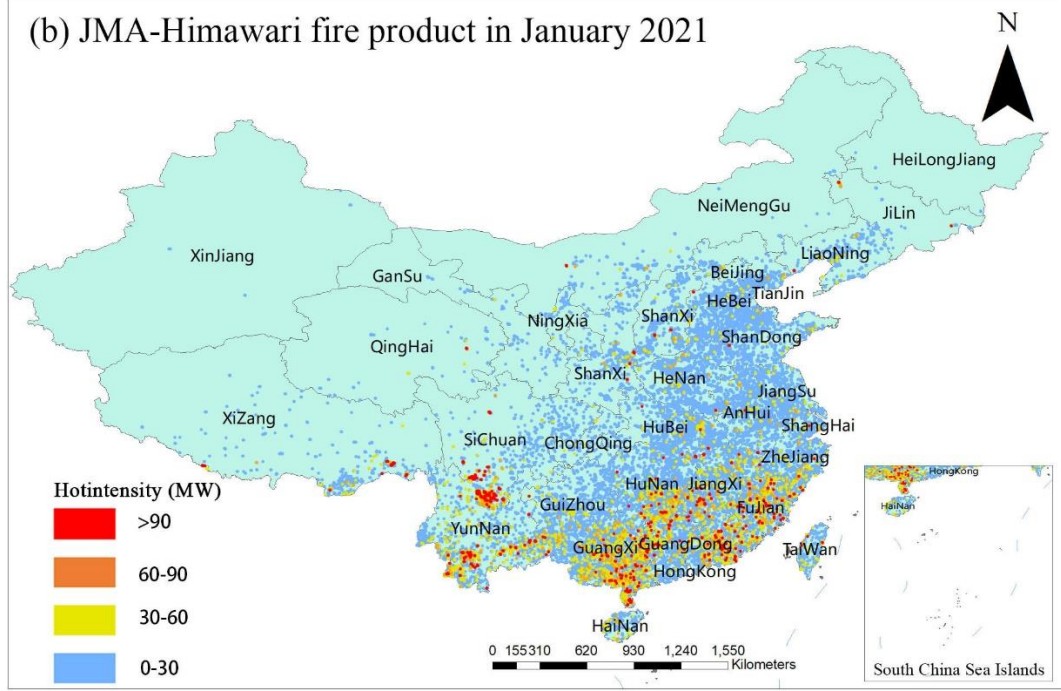

**Figure 1: A brief illustration of JMA- Himawari and NSMC-Himawari fire product in January 2021.**



## 3 Methods

This section mainly introduces our automatic fire-identification algorithm for producing NSMC-Himawari fire product, based on original data obtained from Himawari-8/AHI.

### 3.1 Principle of fire-pixel identification

The basic principle of fire detection based on remote sensing mainly depends on two conditions: the increase of fire temperature leads to the enhancement of thermal radiation, and the difference in the growth range of radiation energy between different thermal infrared bands (Csiszar et al., 2006). Different objects in nature have a diversity of spectral characteristics due to their special temperature and physical and chemical properties (Herold and Roberts, 2005). When biomass is burned, the main radiation sources are flame, carbides with higher temperature, water vapor, smoke and so forth

(Zhang and Kondragunta, 2008). According to Stephen-Boltzmann formula (Quinn and Martin, 1985), notable variations of surface radiation is sensitive to the temperature, which means the heat sources can be detected based on radiance change.

$$J = \varepsilon \sigma T^4 \tag{1}$$

Where $J$ is radiant emittance. $\varepsilon$ is the radiant coefficient. If it is an absolutely black body, $\varepsilon = 1$. $\sigma$ is Stefan-Boltzmann constant, $\sigma = 5.67 \times 10\text{-}8$ W·m⁻²·K⁻⁴. $T$ is absolute temperature.

Wien's displacement formula shows that the brightness temperature is inversely proportional to the wavelength of the

radiation center. With the increase of brightness temperature, the radiation wavelength becomes shorter (Kuenzer and Dech, 2013).

$$\lambda_{max} = \frac{b}{T} \tag{2}$$

Where $\lambda_{max}$ is the peak wavelength of radiation (unit: m). b is the proportional constant, Wien displacement constant, and the value is 0.002897 m·K. $T$ is the absolute temperature. Blackbody temperature $T$ is inversely proportional to peak radiation wavelength $\lambda_{max}$, as the higher temperature can lead to the smaller peak radiation wavelength.

The peak wavelength of surface radiation within normal temperature is close to that at Band 13 and Band 14. When biomass is burned, the temperature can reach 750 K, and the radiation peak wavelength is close to Band 7 (Yang et al., 2017). Taking the radiation when the surface features are not burning as the background radiation, the difference between the combustion radiation and the background radiation is notably in fire pixels, based on which the fire information can be extracted and analyzed. Band 7 of Himawari−8/AHI is mid-infrared, with a wavelength of 3.88 μm, while Band 13 is far-infrared, with a

wavelength of 10.41 μm, and Band 14 is infrared split-window band, with a wavelength of 11.23 μm, respectively. For this research, we considered Band 7, Band 13 and Band 14 for fire-pixel extraction.

According to the observation characteristics of geostationary satellites and the unique types of regional underlying surface in China, the fire-identification algorithm for producing NSMC-Himawari fire product has been further improved and enhanced, from our FY-3D global fire-identification algorithm (Chen et al., 2022a). The major improvement of this fire-extraction
algorithm, compared with previous algorithms, lies in the dynamic adjustment of fire-identification thresholds and the reprocessing of the heat source. The general steps for generating NSMC-Himawari fire product is briefly explained in Fig. 2, and the significant procedures are introduced as follows.

**Figure 2: General flowchart for generating NSMC-Himawari fire products.**

### 3.2 The fire-identification algorithm of NSMC- Himawari fire product

**3.2.1 Detection of cloud pixels**

The cloud coverage has a major impact on the confidence level of fire products (Schroeder et al., 2008). On one hand, the existence of cloud can easily block the ground fire information, causing omission and commission errors. On the other hand, the specular reflection of cloud may lead to the misjudgment of fire pixels (Arino et al., 2012). Therefore, cloud detection is

an important step for effective identifying fire pixels. Wildfires in China are mainly concentrated in the Northeast, North,

South and Southwest China, with differences across months (Cui et al., 2019).

According to the distinct geographical and climatic characteristics, the fire prevention period, which is decided by the State Forestry Administration in China, is mainly divided into autumn-winter and winter-spring. However, from June to September, the number of fire spots decreases significantly, owing to ample rainfall, higher humidity of combustible substance and wider cloud coverage.

To better present the influence of cloud, we displayed the FY-4A cloud coverage data in 2020, including 75000 cloud images, to briefly demonstrate the spatial distribution of cloud coverage (as shown in Fig. 3). From May, the cloud coverage of the entire country gradually increased, and the cloud coverage in other areas except Sichuan Basin decreased from November to next February. Generally, under the same ground and observation conditions, the higher cloud coverage, the less number of fire pixels identified by satellite.

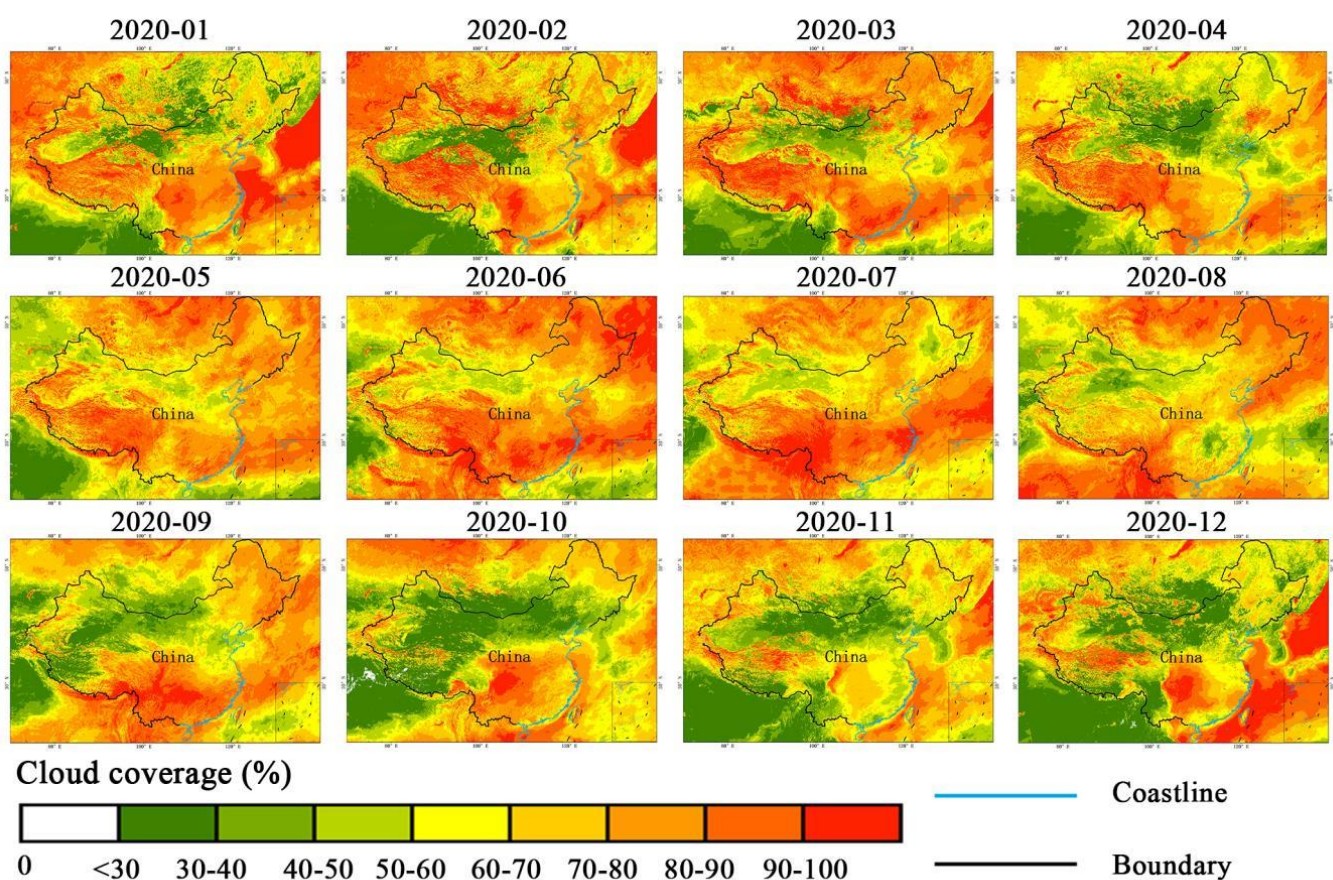

**Figure 3: The spatial distribution of cloud coverage across China in 2020 (based on the Fengyun-4a satellite data).**



The principle of cloud-identification for this research is similar to that of our previous FY-3D fire products (Chen et al., 2022a). According to the difference of brightness temperature and reflectivity between cloud and ground pixels, high-reflection characteristics of cloud pixels in visible-light bands and temperature characteristics in thermal infrared bands were employed for cloud removal. The observed pixels were then divided into cloud pixels and clear sky pixels in different

spectral segments. According to the distribution characteristics of clouds in China, the reflectance difference among cloud, water and land pixels, and our previous product experience of the National Satellite Meteorological Center of China (Chen et al., 2022a, 2022b) we conducted multiple experiments and optimize the threshold value of the cloud-identification algorithm, and re differentiate the criteria. We classified cloud pixels according to the rules shown in Table 2. When the pixel meets any of the following conditions, it was identified as a cloud pixel.

**Table 2. List of cloud pixel determinations.**

| Number | Conditions |
|:---:|:---:|
| 1 | $T_7 - T_{13} < 4$ K |
| 2 | $T_7 - T_{13} > 20$ K & ($T_7 < 275$ K \| $T_{13} < 270$ K) |
| 3 | $R_{vis} > 0.28$ & $\theta_{sz} < 70°$ |
| 4 | $T_{14} < 265$ K |
| 5 | $T_{13} < 270$ K & ($T_{13} - T_{14} < 4$ K \| $T_{13} - T_{14} > 60$ K) |

$T_7$ is the brightness temperature at mid-infrared band; $T_{13}$ is the brightness temperature at far-infrared band; $T_{14}$ is the brightness temperature at far-infrared band; $R_{vis}$ is the reflectivity at visible light band; $\theta_{sz}$ is solar zenith angle. Note: these five rules are set to exclude a diversity of cloud biases and a pixel that meets any rule in Table 2.

### 3.2.2 Calculation of background temperature

As a generally accepted principle for fire detection, if there is a true fire spot in the observation pixel, the brightness temperature of the pixel at Band 7 and 14 is usually notably higher than the background brightness temperature. And the temperature difference at Band 7 is also higher than that at Band 14. In this case, the brightness temperature of background pixels is an important physical index for automatic fire-identification in our research. As the background temperature cannot be obtained from the fire pixels, it should be calculated according to the average of their surrounding pixels. However, if the

average brightness temperature of surrounding pixels selected is excessively high, there may be massive omission errors in fire detection. Conversely, if the average of surrounding pixels selected is lower than normal background brightness temperature, a great number of commission errors may be generated. Therefore, appropriate selection of surrounding pixels is essential for calculating brightness temperature of background pixels. When calculating the background brightness temperature, pixels that may contain fire pixels, water pixels, cloudy pixels and other pixels influenced by solar flare, with





excessively low and high brightness temperature, should be regarded as invalid pixels and removed before background temperature calculation.

According to repeated experiments, based on the windowing method, we selected TOP 20 % high temperature pixels in the initial window area (7 × 7) as the suspected high-temperature, and further identified and eliminated them gradually. The suspicious high-temperature pixels can successively be determined, if their brightness temperature at Band 7 meets the

following equation:

$$T_7 \geq T_{13} + 100 \times R_{vis} + 20K \tag{3}$$

Where $T_7$ is the brightness temperature of the suspicious high-temperature pixel at Band 7. $T_{13}$ is the brightness temperature of suspicious high-temperature pixel at Band 13. $R_{vis}$ is the reflectivity value of suspicious high-temperature pixel in the visible light band.

After removed the above mentioned disturbing pixels in the window area, we calculated the background temperature based

on the remained effective background pixels around the candidate fire pixel. Specifically, the background brightness temperature, the averaged brightness temperature of the effective background pixels at Band 7 in the window area, could be calculated through the following method.

$$T_{7\_bg} = (\sum_{i=1}^{n} T_{7i})/n \tag{4}$$

Where $T_{7\_bg}$ is the averaged brightness temperature of effective background pixels at Band 7 the in window area, which represents the background brightness temperature. $T_{7i}$ is brightness temperature of effective background pixels at Band 7. n

is the number of effective background pixels.

Under clear sky conditions, the satellite could better monitor the fire information on the underlying surface, and the window size was set to 7×7 pixels as initial window area. If the number of available monitored pixels in the window was less than 20 % of the total pixels in the neighborhood due to cloud cover and other factors, then the 7 × 7 window was extended to 9 × 9, 11 × 11 and 19 × 19. If it remained not applicable, these candidate fire pixels were directly marked as non-fire pixels.

**3.2.3 The identification of fire pixels**

The geostationary meteorological satellite can access 24-hour observation. Since the monitored fire characteristics are changeable, it is necessary to dynamically adjust the identification threshold considering the influence of various factors. The main factors include the solar altitude angle, cloud surface and the zone bare of vegetation, which have a great impact on the discrimination accuracy. For instance, the solar altitude angle affects the brightness temperature of the detection area. The

solar altitude angle of each detection area increases from 0° to the maximum angle with time, and then decrease to 0°. The solar radiation intensity received by the surface increases with the increase of solar altitude angle, and the increment of brightness temperature is different in different regions. If we employ a fixed identification-threshold in all different regions in China, it inevitably causes a large number of omission and commission errors. At the same time, the capability of non-



vegetation areas to absorb solar radiation is stronger than that of vegetation, and the difference of brightness temperature

between such non-vegetation pixels and surrounding pixels is significant. Therefore, to increase the accuracy of fire detection, it is essential to dynamically adjust background coefficients according to solar altitude angle, cloud coverage and non-vegetation pixels. Based on the review of historical fire records, we set the identification thresholds and correction coefficients, which varied across monitoring time and observation areas. According to the following condition-check (absolute and relative conditions), fire pixels could be effectively extracted. When the following conditions are met, a pixel

can be identified as a fire pixel:

$$T_7 > 360K \ and \ R_{vis} < 0.7 \ and \ \theta_{sz} > 87° \tag{5}$$

$$T_7 \geq T_{7\_bg} + a(P_v, P_c, \theta_{sz}) \times \delta T_{7\_bg} \ and \ T_{7\_13} \geq T_{7\_13bg} + a(P_v, P_c, \theta_{sz}) \times \delta T_{7\_13\_bg} \tag{6}$$

Where $T_7$ is the brightness temperature of the candidate pixel at Band 7. $R_{vis}$ is the reflectivity of the identified pixel at the visible-light band. $\theta_{sz}$ is solar zenith angle. $T_{7\_bg}$ is the background brightness temperature. $T_{7\_13}$ is the difference of brightness temperature between Band 7 and Band 13. $T_{7\_13\_bg}$ is the difference of background brightness temperature between Band 7 and Band 13. This condition is set to identify the difference under a variety of underlying surfaces in the

window. $\delta T_{7\_bg}$ is the standard deviation of the brightness temperature of background pixels. $\delta T_{7\_13\_bg}$ is the difference of the standard deviation of background brightness temperature between Band 7 and Band 13.

The specific calculation method of the standard deviation is as follows :

$$\delta T_{7\_bg} = \sqrt{\sum_{i=1}^{n} (T_{7i} - T_{7\_bg})^2 \big/ n} \tag{7}$$

$$\delta T_{7\_13\_bg} = \sqrt{\sum_{i=1}^{n} (T_{7\_13i} - T_{7\_13\_bg})^2 \big/ n} \tag{8}$$

Where $n$ is the number of effective background pixels in the window. $T_{7i}$ is brightness temperature of effective background pixels at Band 7. $T_{7\_13i}$ is the difference of brightness temperature of effective background pixels between Band 7 and Band

13. When the land cover types in the window are generally consistent, $\delta T_{7\_13\_bg}$ is relatively small. For the candidate fire pixels, when $\delta T_{7\_13\_bg}$ is smaller than 2 K, the value is set to 2 K. When $\delta T_{7\_13\_bg}$ is large than 4 K, the value is replaced by 4 K.

In the identification algorithm, $a(P_v, P_c, \theta_{sz})$ is a coefficient function, which is related to the non-vegetation pixel ratio, cloud pixel ratio and solar zenith angle. The coefficient varies with different monitoring areas, detection time and angles. Therefore,



the identification threshold can be dynamically adjusted according to different situations. The calculation method is as follows:

$$a(P_v, P_c, \theta_{sz}) = \begin{cases} (sin\theta_{sz} + 1) \times (1 + P_v) \times (1 + P_c); & \theta_{sz} < 60° \\ (1.2 * sin\theta_{sz} + 1) \times (1 + P_v) \times (1 + P_c)^2; & \theta_{sz} \geq 60° \end{cases} \tag{9}$$

Where $P_v$ is the proportion of non-vegetation pixels in window. $\theta_{sz}$ is the solar altitude angle of the identified pixel. $a(P_v, P_c, \theta_{sz})$ increases with $\theta_{sz}$ and $P_v$ to reduce the false errors generated by the solar reflection signal. $P_c$ is the proportion of cloud pixels in window area, and the use of $P_c$ can reduce the false identification of fire pixels at the edge of clouds. The

identification threshold increases with $P_c$.

### 3.3 Reprocessing of the fire production

The thermal anomalies signal at the mid-infrared band (Band 7), also includes non-fire heat source information, e.g. the thermal signals caused by cloud edge, high reflection underlying surface, bare ground, artificial buildings, factories and other non-fire heat sources. Therefore, the preliminarily extracted heat sources should be reprocessed to eliminate false fire pixels.

### 3.3.1 The removal of cloud pixels

If extracted high-temperature pixels meet following conditions, it is considered to be affected by cloud and should not be regarded as fire pixels.

$$R_{vis} \geq R_{vis_{bg}} + R_{c\_th} \ \& \ T_{13} \leq T_{13\_bg} - T_{c\_th} \tag{10}$$

Which $R_{c\_th}$ is the discrimination threshold of cloud influence at the visible light, and the initial value is set to 0.15. $T_{c\_th}$ is the discrimination threshold for cloud interference at Band 13, and the initial value is set to 5 K. Rvis is the reflectivity value

of the high-temperature pixel at the visible-light band. $R_{vis_{bg}}$ is the reflectivity of the high-temperature pixel at the visible-light band. $T_{13}$ is the brightness temperature of the high-temperature pixel at Band 13. $T_{13\_bg}$ is the averaged brightness temperature of effective background pixels in the window at Band 13.

### 3.3.2 The removal of the influence of cloud and desert edge

When the pixel is at the edge of the cloud, the brightness temperature of the mixed pixel with a few cloud is lower, as the

cloud temperature is lower than the surface temperature. When calculating the brightness temperature and standard deviation of background pixels, the mixed pixels are still employed as effective pixels, thus reducing the background brightness temperature and affecting standard deviation. Although cloud pixels have been removed through previous steps, the impact of mixed pixels remains. Based on years of operation and maintenance experience of the National Satellite Meteorological

Center, we have made the following amendments. If the high-temperature pixel meets the following conditions, it is considered to be influenced by cloud and desert edge and not regarded as a fire pixel.

$$T_7 \leq T_{7\_bg} + C \times \delta T_{7\_bg} \ \& \ T_{7\_13} \leq T_{7\_13\_bg} + C \times \delta T_{7\_13\_bg} \tag{11}$$

$C$ is the recognition threshold of cloud edge and desert edge, and the initial value is set to 8. $T_{7\_13}$ is the difference of the brightness temperature between Band 7 and Band 13. $T_{7\_13\_bg}$ is the difference of background brightness temperature between Band 7 and Band 13. $\delta T_{7\_13\_bg}$ is the difference of standard deviation of background brightness temperature between Band 7 and Band 13.

### 3.3.3 The removal of high-reflection underlying surfaces and high-temperature plants

High-temperature factories and high-reflection underlying surfaces are the main ground interference factors affecting the accuracy of satellite-based fire detection. Due to the large amount of thermal radiation generated in the production process, high-temperature factories present similar characteristics to real fire spots at the mid infrared band. For satellite-based fire detection, although these interference sources, which may easily be recognized as false fires, are well known, currently employed fire products rarely exclude these basic thermal anomalies, causing a large uncertain to the product accuracy.

To effectively remove potential false fires and improve the reliability of the Himawari fire products in China, National Satellite Meteorological Centre and State Grid cooperated to generate a data base concerning major high-temperature plants and high-reflection underlying surfaces in China through AI-assisted visual interpretation based on the comprehensive integration of multiple remote sensing sources, including NOAA-20/VIIRS (https://lpdaac.usgs.gov/products/), NPP/VIIRS (https://psl.noaa.gov/data/), TERRA/MODIS and AUQA/MODIS (https://ladsweb.modaps.eosdis.nasa.gov/). The specific parameters of these sources are shown in Table 3.

**Table 3. Main parameters of employed data sources for AI-assisted extraction of heating-sources.**

| Satellite | Sensor | Bands | Resolution (m) |
|---|---|---|---|
| NOAA-20 | | I1 - I5 and M13 | 375 |
| NPP | VIIRS | M5、M7、M11、M13、M15、M16 | 750 |
| TERRA AUQA | MODIS | Band 20 - 36 | 1000 |

The specific procedure for extracting these surface thermal sources is briefly introduced as follows: for interference sources that can generate strong quantity of heat, such as steel plants, thermal plants and chemical plants, we mainly conducted extraction through visual interpretation. The thermal anomalies detected based on NOAA-20, NPP and other remote sensing



sources were matched to Google Maps and further confirmed with high-resolution images. In this case, we identified more than 7000 thermal interference polygons and added to the database.

Since the observation angle of the geostationary meteorological satellite is fixed, the impact of ground mirror reflection on the observation instrument is a common cause for falsely detected fires. Specifically, large photovoltaic panels in

photovoltaic power plants, which are wildly distributed, are the major sources influencing the extraction of actual fires. Firstly, we manually extracted a large number of photovoltaic panel images from the Google satellite images, and employed them as a sample to conduct sample training through deep learning, based on which an extraction model was established; Secondly, we employed an image segmentation-classification model (Yuan et al., 2013) to conduct a large-scale extraction of photovoltaic panels in China, and established a primary sample library of photovoltaic panel. Thirdly, massive manually

editing was conducted to maximally reduce the classification errors. Successively, part of the manual identification results were added to the training samples to further improve the model accuracy. Based on this strategy, we classified more than 9900 photovoltaic panel pixels. The minimum area of extracted photovoltaic panels was dozens of square meters, and the maximum was dozens of square kilometers (as shown is Fig. 4).

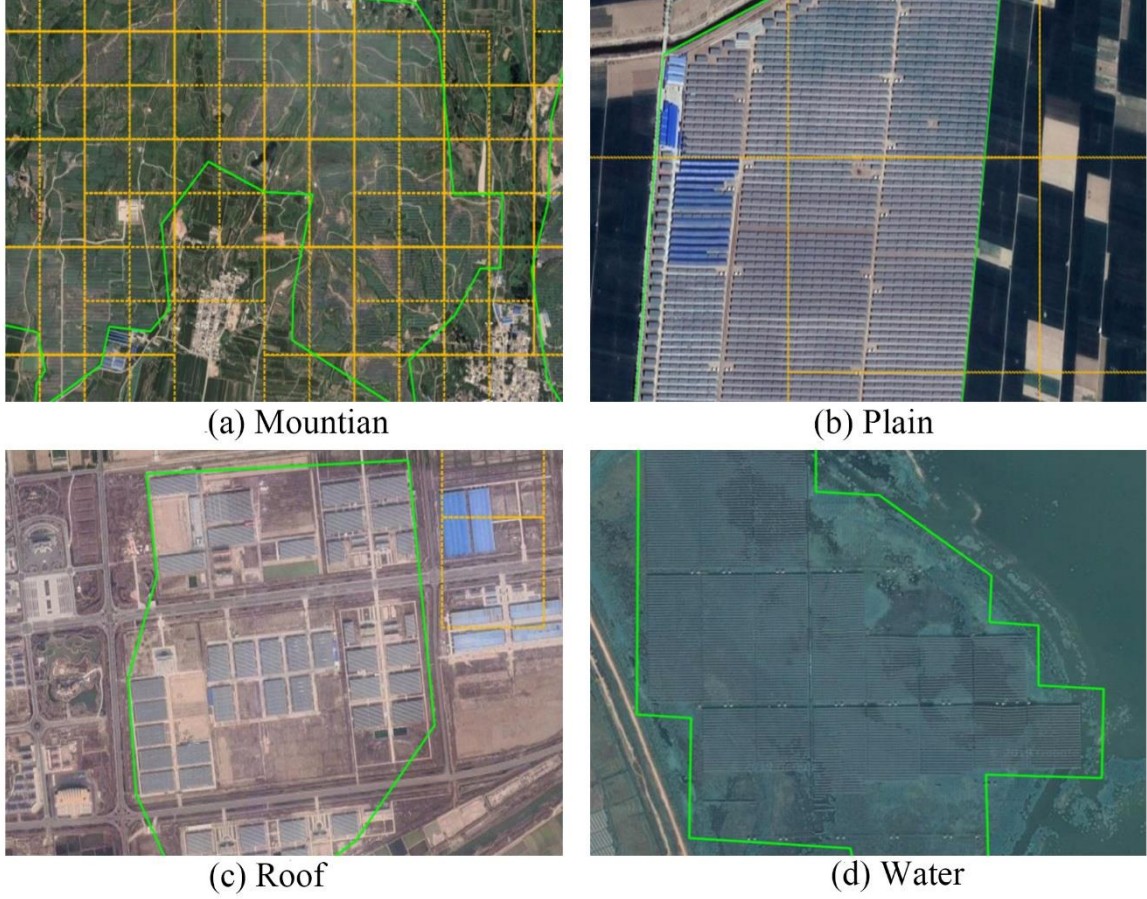

(a) Mountian      (b) Plain

(c) Roof      (d) Water

**Figure 4: Samples of extracted photovoltaic panels (aerial images from © Google Maps).**


Based on accuracy assessment, the primary accuracy of AI recognition is about 87 %. With the following manual editing, the overall accuracy of the extracted thermal sources and photovoltaic panels exceeded 95 %. This dataset of high-temperature plants and high-reflection underlying surfaces (as shown in Fig. 5) has been updated annually by China Meteorological Administration and State Grid. By comparing pre-selected fire pixels and this data set, we removed a majority of false fire

pixels and largely enhanced the accuracy of our NSMC-Himawari fire product.

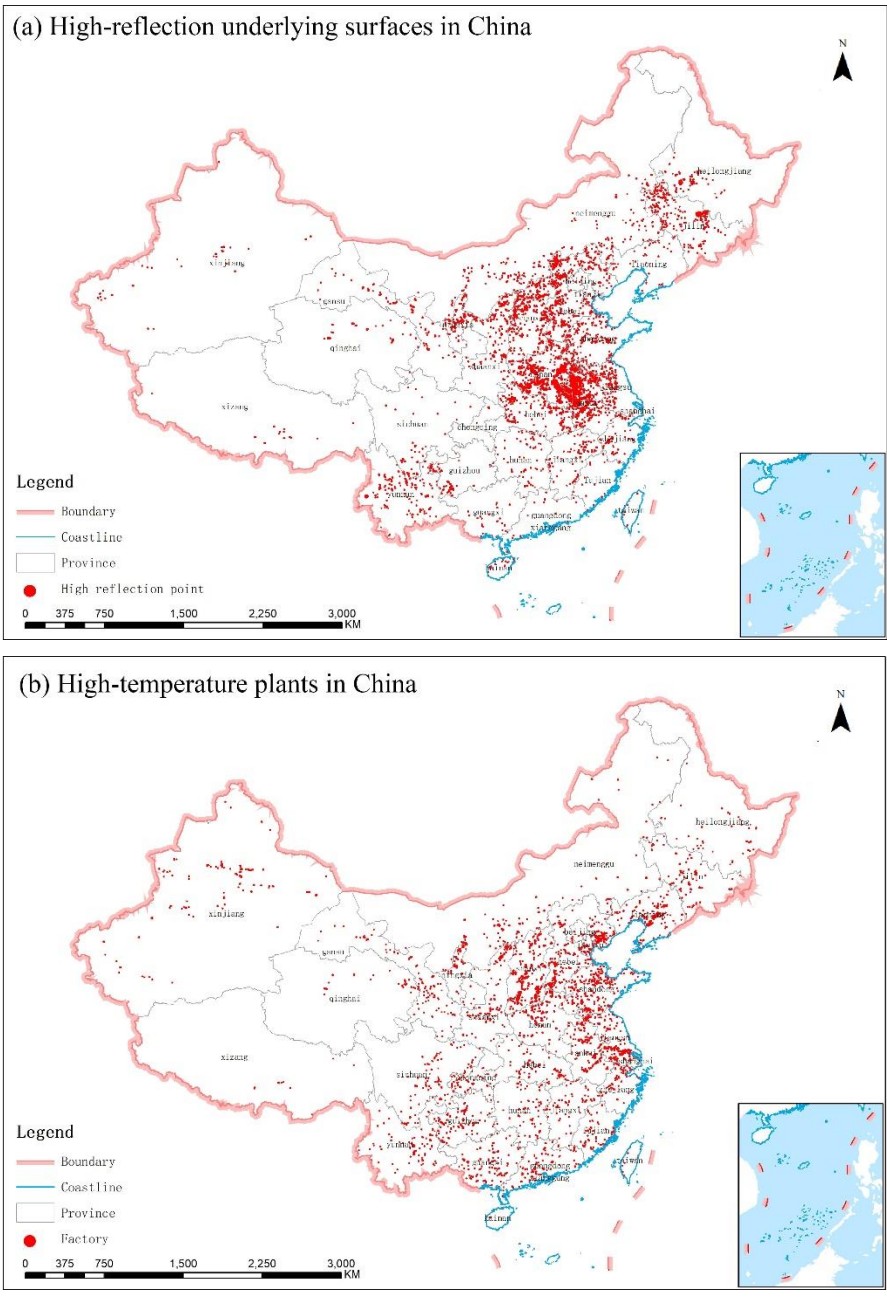

**Figure 5: The high-temperature plants and high-reflection underlying surfaces identified in China for 2021.**



### 3.4 Verification

Our NSMC-Himawari fire product was evaluated using the JMA-Himawari fire products and on-site collected reference of true fires. Specifically, the evaluation was conducted in the following steps. Firstly, through excessive manual verification,

we compared the number of ground interference sources (high-temperature plants and high-reflection underlying surfaces), which are the main source of falsely detected fire pixels, included in the JMA-Himawari and NSMC-Himawari fire products. Secondly,we examined the consistency of NSMC-Himawari and JMA-Himawari fire product in China from the perspective of the total number of fire pixels, different underlying surfaces and different brightness temperatures. Finally,we employed a rare dataset of on-site collected burning records, which was jointly established by the State Grid and China

Meteorological Administration throughout 2021 in five provinces (Guangdong, Guangxi, Yunnan, Guizhou and Hainan) in China, as reference data to evaluate the overall accuracy of NSMC-Himawari fire products in China.

### 4 Result

#### 4.1 Accuracy assessment of JMA-Himawari and NSMC-Himawari fire products in terms of falsely identified interference sources

Based on the original hourly Himawari-8 data from 2019 to 2021, we produced NSMC-Himawari fire product through above introduced fire-extraction approach. Through excessive manual one-by-one-pixel check, we compared the number of interference sources misidentified in the JMA-Himawari and NSMC-Himawari fire product.

As shown in Fig 6 (a), the number of misidentified fire pixels in JMA-Himawari product, majorly caused by ground thermal sources, is much larger than NSMC-Himawari product in all months. The monthly misidentified fire pixels for JMA -

Himawari fire product was on average 13000, ranging from 5000 to 20000 in different months, making the number of hourly misidentified fire pixels close to 18. As a comparison, the monthly misidentified fire pixels for NSMC-Himawari fire product was on average less than 1500, ranging from 200 to 4500 in different months, making the number of hourly misidentified fire pixels close to 2.

To further present the influence of interference sources in the two products, we also calculated the ratio of monthly true fire

pixels to misidentified pixels. As shown in Fig. 6 (b), except from June to September, the true/misidentified ratio for NSMC-Himawari fire product is notably higher than that of JMA. The largest ratio value 58 appeared in April that is, there was only one interference source in every 59 identified pixels. From May to September, due to the relatively small number of real fire pixels, the true/misidentified ratio for both products was smaller than 10, with a minimum value close to 1:1 in August, suggesting the influence of interference sources was extremely strong influence in less-fire seasons.


**Figure 6: The influence of interference sources in JMA-Himawari and NSMC-Himawari fire products across months. (a) The total number of misidentified fire pixels. (b) The average ratio of true to misclassified fire pixels.**

## 4.2 Consistency between JMA-Himawari and NSMC-Himawari fire product

In addition to accuracy assessment in terms of misidentified fire pixels, we also examined the consistence between JMA-

Himawari and NSMC-Himawari fire product in terms of the number of fire pixels, underlying surfaces and brightness
temperature.

### 4.2.1 Consistency in terms of the total number of fire pixels

From January 2019 to December 2021, there were totally 1,136,119 fire pixels identified in NSMC-Himawari fire product while there were 3,232,940 fire pixels identified in JMA-Himawari fire product. The total number of identified fire pixels

from JMA-Himawari fire product was around three times of that of NSMC-Himawari fire product. As introduced above, the large difference was majorly attributed to the fact that NSMC-Himawari fire product avoided a majority of ground interference thermal sources.

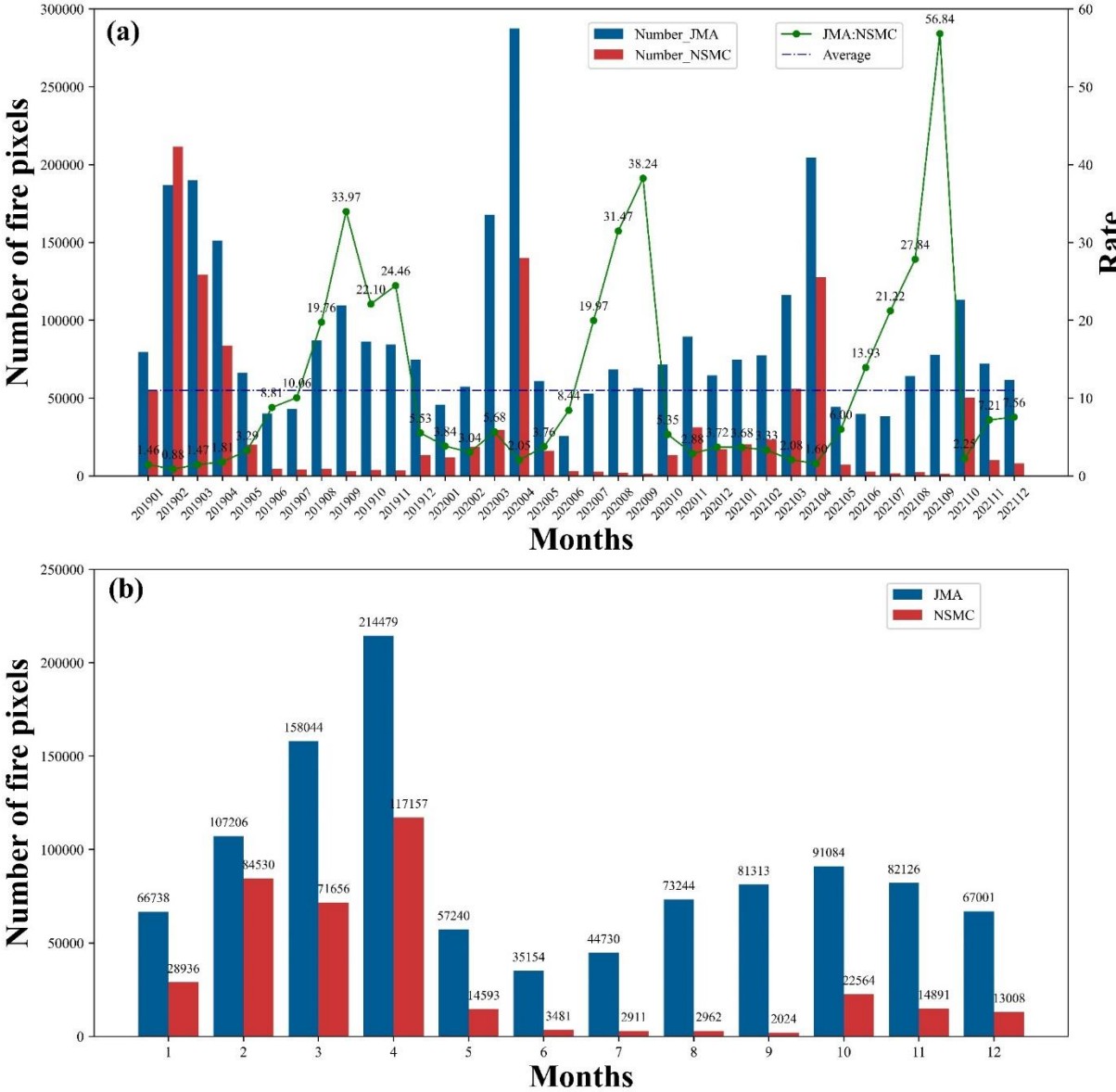

**Figure 7: The total number of identified fire pixels in two fire products. (a) Time series. (b) Monthly average.**



At the monthly basis (Fig. 7), For NSMC-Himawari fire product, identified fire pixels majorly concentrated from January to April while the number of fire pixels was small from June to September every year. As a comparison, for JMA-Himawari fire product, in addition to the period from January to April, there was another peak fire season from June to September. The number of identified fire pixels in JMA-Himawari product was on average ten times of that of NSMC-Himawari fire product from June to September, which is actually a season with a relatively small number of forest fires and crop-residue burning.

This strong inconsistence further proved the importance of removal of potential interference fire pixels and the reliability of NSMC-Himawari fire product, especially in the less-fire seasons.

We further examined the overall consistence between JMA-Himawari and NSMC-Himawari fire products from two different perspectives. From the JMA perspective, we mainly checked the average possibility of one identified fire pixel in JMA-Himawari fire product also included in the NSMC-Himawari fire product. From the NSMC perspective, we mainly checked

the average possibility of one identified fire pixel also included in the JMA-Himawari fire product.

As shown in Fig. 8, the consistency was relatively low from the period from June to September, and much higher from the period from January to April. The consistence from the NSMC perspective, ranging from 42 % to 87 % across months, was much higher than the consistence from the JMA perspective, ranging from 1 % to 62 %. This is majorly attributed to the large amount of false fire pixels included in the JMA perspective have been removed in NSMC-Himawari fire product, and

thus the possibility of finding a counterpart was largely reduced. Meanwhile, JMA-Himawari fire product retained a majority of effective true fire pixels in the original Himawari files and thus the possibility of finding a counterpart in JMA-Himawari fire product was relatively large. The consistence check further proved that NSMC-Himawari fire product effectively maintained useful information from original Himawari sources and presented a largely enhanced reliability.

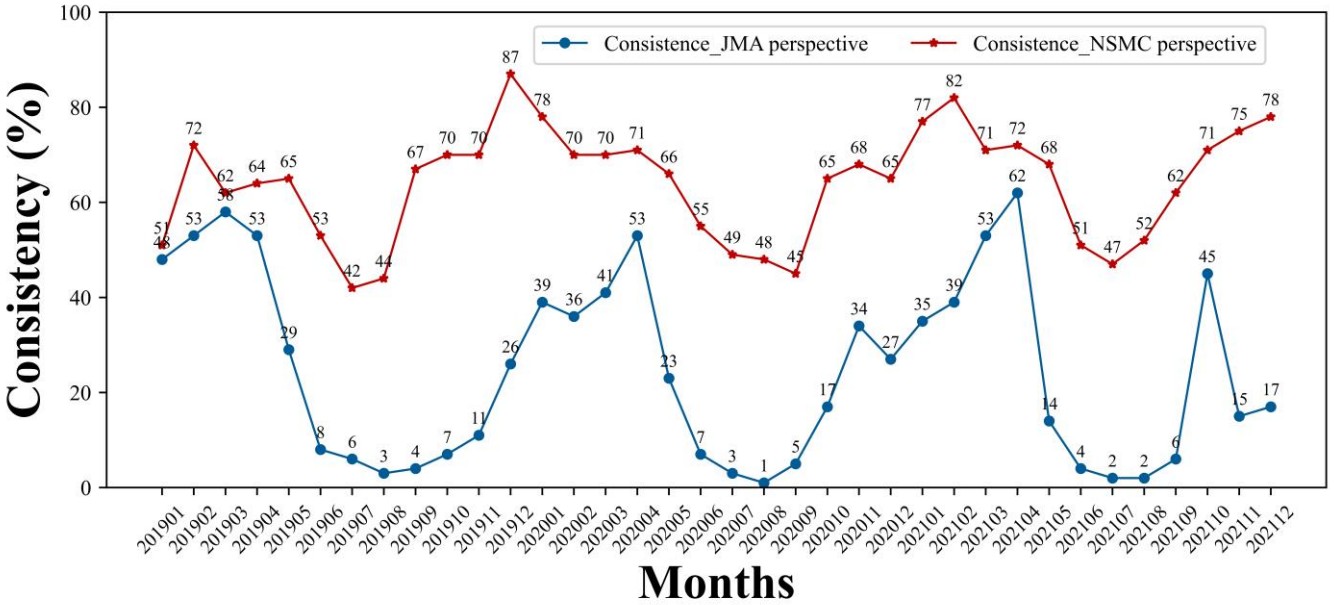





**Figure 8: The general consistency between JMA-Himawari and NSMC-Himawari fire product across months.**

**4.2.2 Consistency on different underlying surfaces**

China has a vast territory and a diversity of underlying surfaces, leading to large variations of fire types and difficulty of fire detection across regions. Specifically, compared with southern forest wildfires that could generally burn very large areas and last several months, fires in the Northeast of China are majorly small-sized fires that burn in relatively low intensity in agriculture and forestry lands. According to Climate Change Initiative (CCI) Land Cover V2

(http://maps.elie.ucl.ac.be/CCI/viewer/download.php), with a spatial resolution of 300 m in 2015, We divided the underlying surfaces into Built-up areas, Farmland (straw), Grass (grassland) and Forest, respectively.

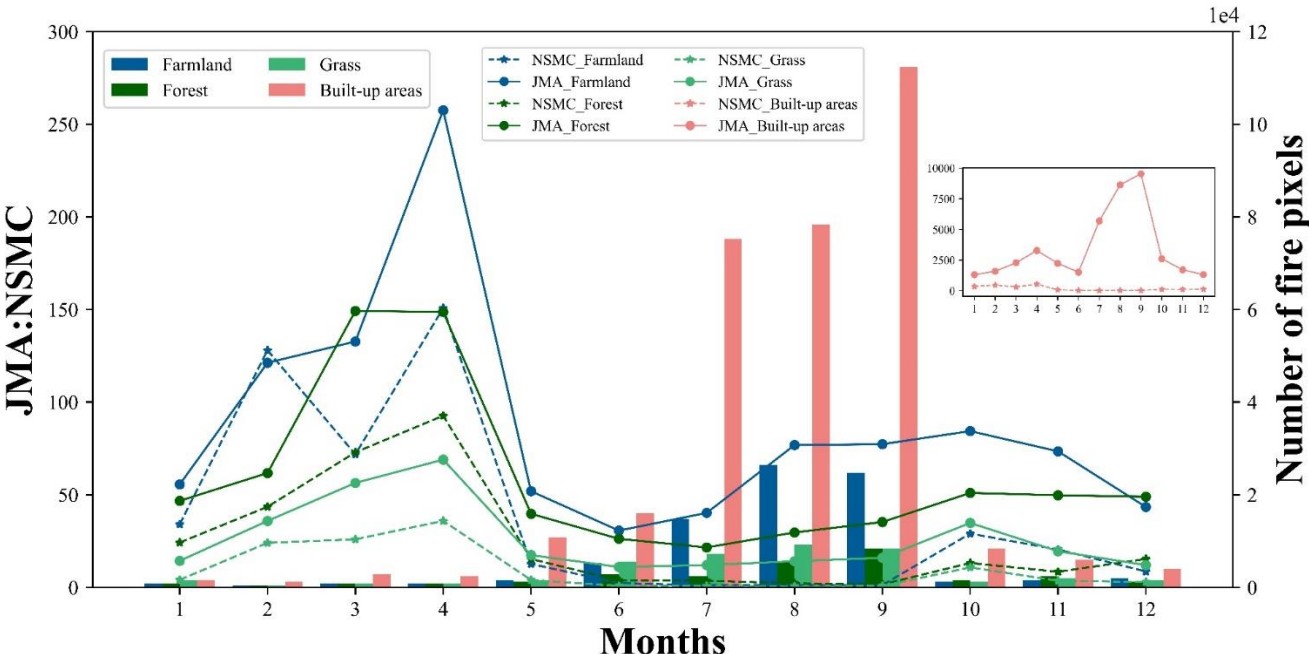

**Figure 9: The number of fire pixels and ratio between JMA and NSMC fire products under different underlying surfaces across months.**

Figure 9 shows the monthly average of fire pixels identified on different underlying surfaces from 2019 to 2021. The total number of fire pixels in farmland was the largest for both fire products, followed by forest, and the general trend of the two fire products was similar in number. Generally, as Table 4 shown, the consistence between two products from NSMC perspective was relatively high for all underlying surfaces, with an average consistence larger than 70 %. As a comparison, from JMA perspective, the consistence between two products was relatively low for all underlying surface types. Figure 9

also suggested that the number of fire pixels identified on building areas in JMA-Himawari fire product during June to September was more-than-100 times of that identified in NSMC-Himawari fire product. Given that there is actually a very



limited number of fires occurred on buildings during this period, it is highly possible the excessive number of fire pixels detected on building areas in JMA-Himawari fire product resulted from false alarms caused by ground thermal sources. Meanwhile, the maximally reduced fire-pixels identified on building pixels additionally proved the advantage of employing our ground-thermal-source database to remove false alarms.

**Table 4. The consistence between JMA and NSMC-Himawari fire product under different underlying surfaces.**

|  | Jan (%) | Feb (%) | Mar (%) | Apr (%) | May (%) | Jun (%) | Jul (%) | Aug (%) | Sep (%) | Oct (%) | Nov (%) | Dec (%) |
|---|---|---|---|---|---|---|---|---|---|---|---|---|
| NSMC_Farmland | 63.50 | 64.43 | 62.18 | 68.75 | 63.64 | 55.51 | 45.20 | 48.60 | 64.30 | 71.31 | 67.07 | 70.27 |
| JMA_Farmland | 42.06 | 48.54 | 40.68 | 55.78 | 19.16 | 5.10 | 1.84 | 0.87 | 1.43 | 25.09 | 21.43 | 18.14 |
| NSMC_Forest | 73.58 | 74.43 | 74.51 | 70.17 | 70.69 | 57.76 | 49.98 | 51.92 | 65.12 | 70.56 | 79.54 | 81.38 |
| JMA_Forest | 47.51 | 53.37 | 51.62 | 60.68 | 32.85 | 10.07 | 10.06 | 4.51 | 4.17 | 22.10 | 18.87 | 35.96 |
| NSMC_Grass | 69.10 | 67.44 | 63.04 | 64.22 | 60.99 | 39.41 | 38.42 | 44.66 | 64.42 | 67.14 | 67.18 | 71.67 |
| JMA_Grass | 28.62 | 45.90 | 38.91 | 52.51 | 17.97 | 3.22 | 2.97 | 2.50 | 4.54 | 25.48 | 14.14 | 23.77 |
| NSMC_Built_up areas | 79.47 | 71.02 | 73.02 | 67.50 | 68.11 | 63.89 | 51.62 | 47.65 | 73.33 | 78.14 | 83.81 | 90.03 |
| JMA_Built_up areas | 15.23 | 14.76 | 8.11 | 14.10 | 2.96 | 1.10 | 0.35 | 0.21 | 0.23 | 5.42 | 4.31 | 4.77 |

### 4.2.3 Consistency under different brightness temperature

The key to extract fire pixels is to precisely obtain their brightness temperature and set fire-extraction thresholds accordingly. Consequently, the brightness temperature of identified fire pixels has a notable influence on the consistence between JMA-Himawari and NSMC-Himawari fire product.

As shown in Fig. 10, the higher the brightness temperature, the higher consistence was. When the brightness temperature is less than 270 K, the consistency between the two fire products was less than 20 %, and the consistency increased rapidly with the rise of the brightness temperature. When the brightness temperature exceeded 330 K, the consistency between two products exceeded 70%. It is noted here that the brightness temperature for a fire pixel was not solely decided by the temperature of fires. Since a occurring fire sometimes simply occupied a very small percentage of a 2 km × 2 km cell, then the brightness temperature for this pixel was majorly decided by the background brightness temperature. When the brightness temperature was extremely high (e.g. over 330 K) and exceeded normal background temperature, this pixel was most likely a true fire spot, which thus was included by both the JMA-Himawari and NSMC-Himawari fire product. When the brightness temperature of one pixel was relatively low (e.g. around 270 K), yet much higher than its surrounding pixels,



it may also be considered as fire pixel. This phenomenon is common for fire detection in northern China in winter. However, the existence of ground thermal sources can lead to the misidentification of fire pixels. Since we employed the dataset of ground interference sources, a large amount of misidentified fire pixels under low-temperature scenario, which are included in JMA-Himawari fire product, has been removed from NSMC-Himawari fire product, leading to a very small consistence when the brightness temperature of target pixels was low. This output further revealed the strong influence of ground

thermal sources on the reliance of extracted fire pixels.

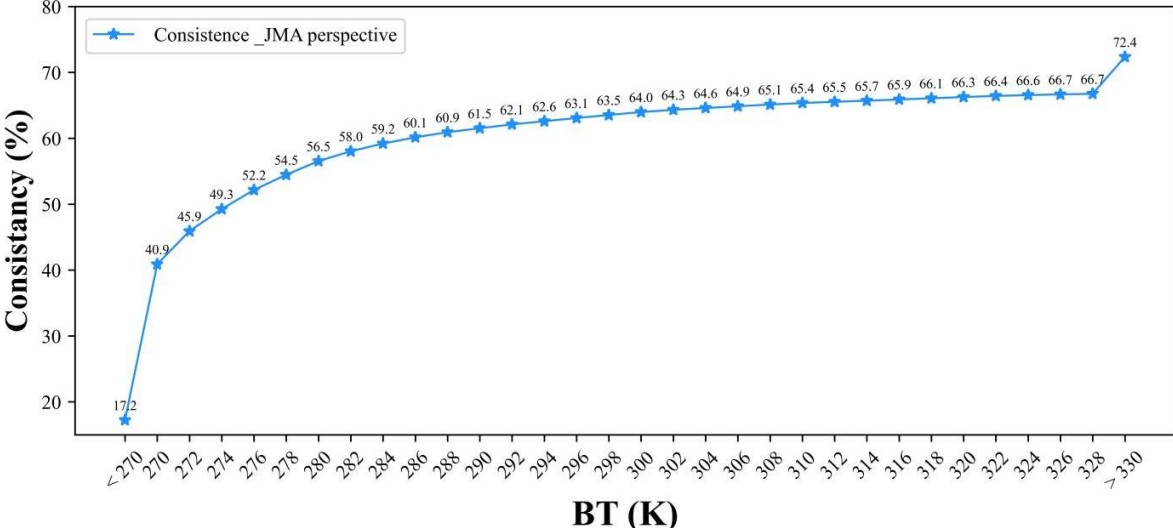

**Figure 10: The variation of consistence between JMA-Himawari and NSMC-Himawari fire product under different brightness temperature.**

### 4.3 Accuracy assessment based on ground truth data

Since the occurrence of wild fires is highly unpredictable and the duration of fires may be short, the real-time capture and continuous monitoring remains highly difficult. Therefore, the major challenge for verifying fire products lies in the lack of ground truth reference, and the reliability of JMA-Himawari fire product in China remains unknown.

To obtain reliable and large scale reference for fire-detection, STATE GRID Corporation of China and China Meteorological Administration jointly conducted annual fire-detection experiment since 2017 in five provinces Guangdong, Guangxi,

Yunnan, Guizhou and Hainan in China. Massive drones have been employed to check the occurrence of fires and report the location and occurring time of actual fires. For this research, we employed field-collected fire reference in the five provinces throughout 2021 for accuracy assessment. By comparing the actual information of fires and the corresponding information in the two products, we could obtained the omission, commission errors and overall accuracy of JMA-Himawari and NSMC-Himawari fire products.

As shown in Table 5 and Fig. 11, there were a proportion of omission errors in both fire products. This type of errors was majorly caused by the limitation of sensors that some small fires within the 2km × 2km grid cannot be identified. Thanks to





the dynamic fire-extraction thresholds adapted to China's local parameters, the number of fire-pixels correctly recognized by NSMC-Himawari fire product (2174) was notably larger than that of JMA-Himawari fire product (1648). Meanwhile, the number of commission errors, which standard for those non-fire pixels misidentified as fire pixels, included in JMA-

Himawari fire product (1160) was much larger than that in NSMC-Himawari fire product (413).

**Table 5. Accuracy assessment of JMA-Himawari and NSMC-Himawari fire products based on field ground truth.**

|  | Correct | Omission errors | Commission errors | Accuracy(%) | Accuracy without omission (%) |
|---|---|---|---|---|---|
| JMA | 1648 | 243 | 1160 | 54 | 59 |
| NSMC | 2174 | 137 | 413 | 80 | 84 |

This is majorly attributed to our reprocessing of the preliminarily extracted fire pixels according to our data base of ground thermal sources, which removed a majority of false fire pixels. Disturbed by a large proportion of misidentified fire pixels, the overall accuracy of JMA-Himawari fire product across China was simply 54 %. Even if we ignore those omission errors

caused by sensor limitations, the overall accuracy was below 60 %. Therefore, the existing JMA-Himawari fire product cannot provide reliable hourly monitoring of wildfires across China.

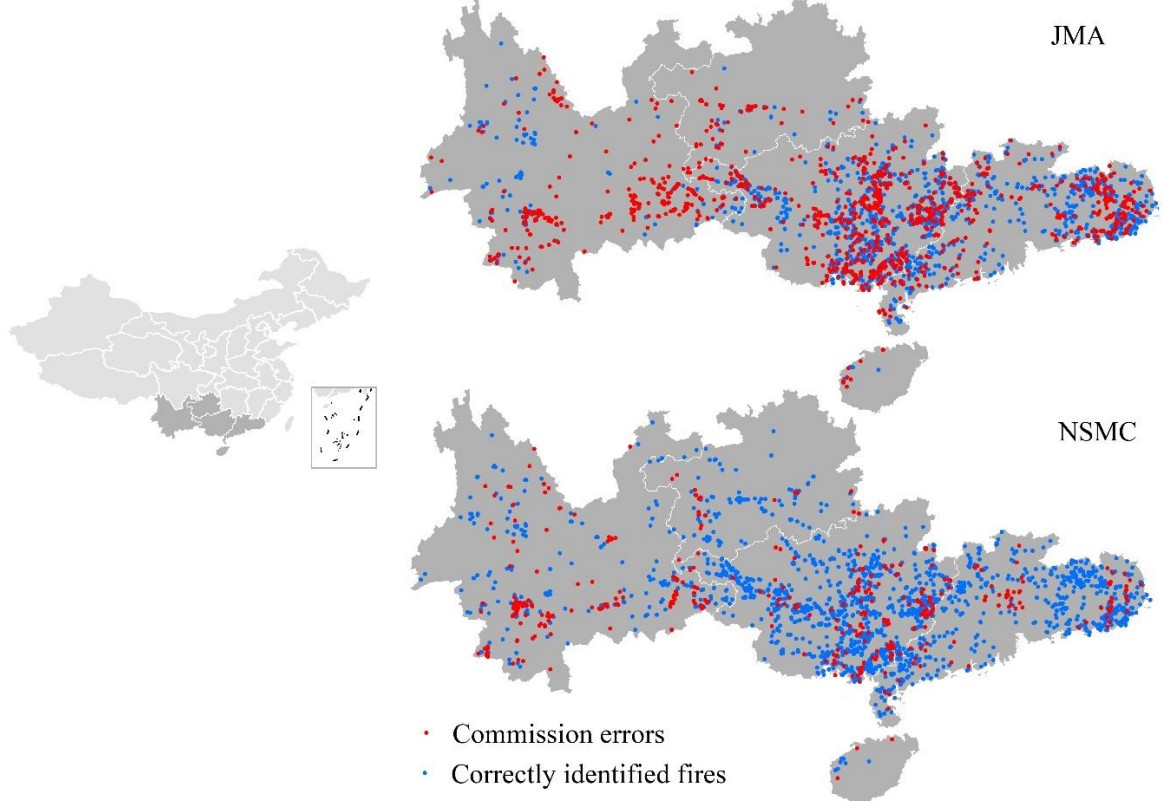



**Figure 11: Accuracy assessment of NSMC-Himawari and JMA-Himawari fire products in China based on the ground-truth data.**

As a comparison, with a notably improved number of correctly identified fire pixels and a significantly reduced number of misidentified fire pixels, the overall accuracy of NSMC-Himawari fire product reached 80 %. If the omission errors were not considered, the overall accuracy of NSMC-Himawari fire product could further reach 84 %. The largely improved accuracy makes our fire product an ideal source for real-time fire monitoring in China.

## 5 Discussion

We compared the accuracy of JMA-Himawari and NSMC-Himawari fire products in China. Based on the field-collected evidence, NSMC-Himawari fire products achieved a notably improved accuracy. This was mainly attributed to the additional consideration of underlying surface conditions, which effectively removed a large proportion of false fires. This result was consistent with the overall accuracy of our previous fire products. By comprehensively consider the underlying surfaces in China, FY-3D global fire product achieved a high consistence with the mainstream and most commonly employed MODIS

fire product at the global scale and significantly improved accuracy in China (Chen et al., 2022). These results suggested that the accuracy of fire products was largely biased by the heating-sources, which cannot be easily avoided through previously set fire-detection thresholds, and additional data sources should be employed for removing this type of major noises. Despite a notably improved accuracy, limitations remains. Firstly, regardless the setting of major fire-detection parameters, the moderate spatial resolution of NSMC-Himawari fire products (2 km) restricted its capability for identifying the occurrence

of those small fires. To address this issue, the comprehensive consideration of NSMC-Himawari fire products and other high resolution fire products (e.g. VIIRS fire products) may be a potential solution. Nevertheless, the reliability of these complementary fire products in China requires further verification and improvement. Secondly, during the early-morning and evening, due to the relatively low solar altitude, the identification of cloud removal was slightly influenced during this period. Thirdly, due to the vast coverage, the conditions of underlying surface varies significantly in China. Therefore, the

setting of fire-identification thresholds may achieve different effects across regions. For those regions with complicated underlying surfaces, the reliability of NSMC-Himawari fire products may be lower than the overall accuracy.

   In recent years, there is a growing need for timely monitoring the occurrence of wild fires in China, especially for the identifying grassland fires and forest fires, which cause severe threat to public health and environment, and detecting human-induced crop-residue burning, which has become one major source for airborne pollutants. However, existing mainstream

fire products cannot fully support these needs. The MODIS and FY-3D daily global fire products can solely visit the target area once per day. Meanwhile, JMA-Himawari hourly fire products have the potential to provide quick-identification of wildfires. However, even without quantitative accuracy assessment, previous implementation conducted by major institutions (e.g. National State GRID and China Meteorological Administration) reported that JMA-Himawari products presented large uncertainties across China. Therefore, JMA-Himawari product was limitedly employed for either scientific

or practical implementations, despite its high temporal resolution. Based on a rare reference data set, this research, for the first time, verified the reliability of JMA-Himawari product, and quantitatively proved that JMA-Himawari product presented a large uncertainty across China and not suitable for real-time fire monitoring. In this case, NSMC-Himawari fire products, with an hourly temporal resolution and an overall accuracy around 80 %, can be an ideal source for monitoring real-time wild fires in China. Furthermore, in recent years, with the increasing strict regulation on crop residue burning,

many farmers choose to secretly burn crop residues in the nighttime, which causes a major challenge for timely monitoring and management. With the NSMC-Himawari fire products, which provides reliable 24h-hourly data sources, automatic monitoring, especially nighttime monitoring, of crop residue burning in China can be effectively implemented.

In previous implementations, scholars are more likely to employ official fire products released by those institutions that are operating corresponding satellites and rarely consider to originally propose a fire product based on the raw satellite data.

This research is a rare attempt to produce an improved fire product based on existing raw satellite source by additionally considering localized complementary sources. The result suggested our NSMC-Himawari fire product achieved a much higher accuracy than the official JMA-Himawari fire product, which is not suitable for practical or research use in China. By analogy, the use of complementary sources (e.g. the location of thermal sources) could further improve the local suitability of other global fire products (e.g. MODIS fire product).

Further improvement of real-time fire detection in China can be explored in the following perspectives. As introduced above, one major disadvantage of Himawari-based fire products is the relative coarse spatial resolution provided by Himawari satellite. As an important part of Fengyun series Meteorological satellites, China plans to launch the new generation of FY-4C stationary satellite with a temporal resolution of 5 minutes and an improved spatial resolution of 1km in 2024, which presents a good potential for largely enhancing the sensitivity of fire identification in China. Meanwhile, time series analysis,

instead of the analysis on individual image, should be additionally considered in the algorithm of fire detection, leading to more effective removal of false fire information. With enhanced resolution and reduced uncertainty, real-time fire can be better monitored in China.

## 6 Conclusions

Given the uncertainty of JMA-Himawari fire product in China, we proposed an adaptive hourly NSMC-Himawari fire

product based on the original Himawari source by employing a dynamical threshold for fire-extraction and a data base of ground thermal sources. Based on the visually-extracted references and consistence check, we found NSMC-Himawari fire product effectively removed a majority of false fire alarms included in the JMA-Himawari fire product. Based on a rare field-collected ground reference dataset, we evaluated the reliability of JMA-Himawari and NSMC-Himawari fire product across China. The overall accuracy of JMA-Himawari fire product was 54 % and 59 % (not considering the omission error)

respectively. As a comparison, by identifying more real fire pixels and avoiding a majority of false fire alarms, the overall accuracy of NSMC-Himawari fire product was 80 % and 84 % (not considering the omission errors) respectively. NSMC-

Himawari fire product can be a promising source for improved real-time fire monitoring across China. This research also provides useful reference for employing local dataset of underlying surfaces and thermal sources to enhance the accuracy of global fire products in specific regions.

**Data availability**

NSMC-Himawari fire product for China are available at http://figshare.com with the identifier doi: https://doi.org/10.6084/m9.figshare.21550248(Chen et al., 2022).

**Author Contribution**

JC, WZ, SW and CL produced NSMC-Himawari fire product. JC, QL and ZC conceived the manuscript. JC, QL, ZC, WZ, YZ, CL and EZ conducted data analysis. JC, QL, XC, JY and QY produced figures. QL and ZC wrote the draft. BG produced the official website. ML, ZC and BG reviewed and revised the manuscript.

**Competing interests**

The authors have no competing interests.

**Financial support**

This research is supported by the National Key Research and Development Program of China (2021YFC3000300) and National Natural Science Foundation of China (Grant No.42171399).

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
