# Peer review of "An adapted hourly Himawari-8 fire product for China: principle, methodology and verification."

_Earth System Science Data, 2022_

## Author Response (AR1)

**To Reviewer 1:**

**Thanks so much for providing us some many general and detailed comments, which helped us so much improve this manuscript. We have fully advised this manuscript according to your constructive comments. We are more than willing to conduct further revisions if you have additional comments.**

**Thanks again for your time and help.**

1. There are some mistakes in grammar in this manuscript. For example, (1) the 1$^{st}$ sentence in Abstract "Wild fires exerts strong influences on the environment, …", (2) Section 3.2.2 "After removed the above mentioned disturbing pixels …"

**R: Thanks so much for pointing this out. We have corrected the grammatical errors accordingly in the revised manuscript.**

2. The Abstract part should concentrate on the adapted hourly Himawari-8 fire product, it is unnecessary to provide the information of JMA-Himawari fire product.

**R: Thanks so much for this comment. Yes, our main work is to generate an adaptive hourly NSMC-Himawari-8 fire product for China based on the original Himawari-8 source. This product has achieved higher accuracy than the original Himawari-8 source (verified by the field-collected ground reference dataset) by adopting the dynamic threshold for fire-extraction algorithm, two special interference source databases, and the reprocessing of fire pixels. Currently, many scholars can only use the original Himawari-8 data from the Japan Meteorological Agency (JMA for short) to improve the temporal resolution when conducting research on wild fires in East Asia. Despite a high temporal resolution, the JMA-Himawari-8 fire product presents large uncertainties in China. Our results also show that the accuracy of NSMC-Himawari-8 fire product is 80 % and 84 % (not considering the omission errors) respectively, much higher than the accuracy of JMA- Himawari-8 fire product , which is 54% and 59%. The reason we put a brief introduction of JMA-Himawari data here is that this is a well-known product from Himawari-8, and if we did not mention JMA-Himawari, readers may be wondering why should we produce a new product based on Himawari-8 when there is one already. So the list the disadvantage of JMA-Himawari fire product, compared with NSMC-Himawari fire products, can help readers better understand the motivation and outputs of this new data product.**

**But your comment is very useful. In the revised manuscript, we have reduced irrelevant introduction and just retained some necessary and brief introduction of JMA-Himawari to highlight the background and major improvement of our new data sources.**

3. There is some redundant information in this manuscript. For example, in the introduction part, the information of "GEOS-16 Advanced Baseline Imager (ABI) fire products" is redundant information, it seems has nothing to do with the manuscript.

**R: Thanks so much for pointing this out. In introduction, we meant to comprehensively summarize the global research progress of real-time monitoring of wild fires. In order to realize real-time monitoring of wild fires, the satellites carried on the sensors with extremely high temporal resolution is required. Both GEOS-16 Advanced Baseline Imager and Himawari−8 Advanced Imager are advanced sensors with high temporal resolution. However, the coverage of GEOS-16 is from 135 ° W to 75 ° W, which can only be used for real-time monitoring of wild fires in the western hemisphere, while Himawari−8 is located over East Asia, which can be used for real-time monitoring of wild fires in China. In this case, GEO ABI and Himawari-8 fire product can meet different requirements of users. Therefore, a brief introduction of GEO ABI fire product can help readers have a big picture of currently available high-temporal-resolution fire products and their characteristics.**

**What you suggested here was very important. In the revised manuscript, we reduced the introduction of GEOS-16 Advanced Baseline Imager (ABI) fire products, and just retained very limited information concerning this data set. Thanks again for your comment.**

4. Figure 3 makes mistakes in the area of nine-dotted line. Cloud coverage in the area of nine-dotted line is only the cloud coverage of background, the actual cloud coverage in the area of nine-dotted line is totally missing.

**R: Thanks so much for pointing this out. Yes, you are right, as we acknowledged, since FY-4A satellite can only monitor about 15 ° N at the southern end of China, it cannot completely cover the all areas within nine-dotted line. Meanwhile, NSMC-Himawari-8 fire product also do not include the small islands within nine-dotted line due to the coarse spatial resolution of Himawari-8, so the missing values of nine-dotted line was not a cause of mistakes. Instead, the missing data was caused by the limitation of FY-4A and Himawari-8.**

**Thanks again for your valuable comments. We have added explanation of the missing data issue to the bottom of the figure capital, so that readers can easily read the reason for the blank area and reproduced the image for a better presentation.**

5. In Table2, the explanation of "$T_7$", "$T_{13}$" and "$T_{14}$" does not mentioned the specific bands, and makes the explanation not clear enough. "$T_{13}$" and "$T_{14}$" even have the same explanation, while they actually not refer to the same temperature.

**R: Thanks so much for pointing this out. We have corrected it accordingly in the revised manuscript. $T_7$ is the brightness temperature at Band 7 (3.89 μm). $T_{13}$ is the brightness temperature at Band 13 (10.41 μm). $T_{14}$ is the brightness temperature at Band 14 (11.24 μm). The specific wavelength of the wave band was explained in advance in Table 1.**

6. Section 3.3.1 and Section 3.3.2 cannot be found in Figure 2. They are important contents in "Data Reprocessing" process, why they are excluded in Figure2? Similarly, Section 3.4 are also excluded in Figure 2, which makes flowchart incomplete.

**R: Thanks so much for this. Yes, our previous flow chart was not complete. According to your suggestion, we have improved and supplemented it in the revised manuscript, including all the significant processing steps (adding the removal of cloud pixels, the removal of the influence of cloud and desert edge and verification).**

**Thanks again for this comment, which help us largely improve this part.**

7. Equation 10 in Section 3.3.1 provides the same explanation for "$R_{vis}$" and "$R_{visbg}$", one of them was given a wrong explanation.

**R: Thanks so much for pointing this out. We are sorry for this typo. $R_{vis}$ is the reflectivity of the high-temperature pixel at the visible-light band. $R_{visbg}$ is the averaged reflectivity of effective background pixels in the window at the visible-light band. We have corrected it accordingly in the revised manuscript.**

**Thanks again for your comment.**

8. Figure 4 gives wrong word "Mountian".

**R: Corrected. Thanks so much for pointing this out.**

9. Section 3.3.3 shows that the accuracy assessment was done for "The removal of high-reflection underlying surfaces and high-temperature plants", however, the uncertainties of other processes are not mentioned in this manuscript. For example, (1) the uncertainty of the "condition-check (absolute and relative conditions)" in Section 3.2.3. (2) the uncertainty of "the identification threshold" in Section 3.2.3. (3) the uncertainty of "the removal of cloud pixels" in Section 3.3.1. (4) the uncertainty of "the removal of the influence of cloud and desert edge" in Section 3.3.2. (5) the representativeness of 5 provinces in Section 3.4. Why the accuracy assessment only done for "The removal of high-reflection underlying surfaces and high-temperature plants"?

**R: Thanks so much for this. This is a very constructive comment and we really should give more details (if the accuracy assessment is possible) or sufficient**

explanation (if the accuracy assessment was not currently available) on the reliability of each step for producing our NSMC-Himawari-8 fire product. In this case, readers can have a better understanding of how to achieve the high-accuracy of this data set.

In fact, as we introduced in Section 3.4, the accuracy verification method we adopted was the analysis of the number of interference sources, the consistency with JMA-Himawari-8 fire product from three aspects, and the most critical dataset of on-site collected burning records in the five southern provinces. In section 3.2.3, 3.3.1 and 3.3.2, you can see that a large number of empirical parameters were used in our methods to ensure the accuracy of identification, but uncertainties cannot be listed one by one. Because the atmospheric composition and underlying surface conditions in China were complex and changeable, and the sources of uncertainty were multiple parameters interacting, it was inexhaustible to use the uncertainties of empirical parameters to evaluate the final results, which was almost impossible. Therefore, the most persuasive method we used was to directly prove the effectiveness of our parameters setting (van der Werf et al., 2017), based on accuracy verification of on-site collected burning records. This was the most direct and persuasive method. In section 4.3, we detailed the verification results, and the final results showed that NSMC-Himawari-8 fire product was highly accurate in China, with the accuracy is 80 % and 84 % (not considering the omission errors), respectively, however, the accuracy of JMA-Himawari-8 fire product is only 54% and 59%, which is the direct evidence to prove the high accuracy of NSMC-Himawari-8 fire product and the effectiveness of our thresholds setting.

In section 3.3.3, we used two unique interference source datasets as auxiliary data (based on AI-assisted visual interpretation) of high-temperature factories and high-reflection underlying surfaces generated by National Satellite Meteorological Centre and State Grid, to remove the false fire alarms in NSMC-Himawari-8 fire product. Because the datasets were not publicly available, we showed the accuracy of the two datasets to let readers know the reliable basis for removing false fire alarms.

Thanks again for the inspiration of your valuable comments. In the revised manuscript, we also showed the accuracy of other auxiliary data (cloud coverage in section 3.2.1).

*van der Werf, G. R., Randerson, J. T., Giglio, L., van Leeuwen, T. T., Chen, Y., Rogers, B. M., Mu, M., van Marle, M. J. E., Morton, D. C., Collatz, G. J., Yokelson, R. J., and Kasibhatla, P. S.: Global fire emissions estimates during 1997–2016, Earth Syst. Sci. Data, 9, 697–720, https://doi.org/10.5194/essd-9-697-2017, 2017.*

10. The left y axis of Figure 6 (a) actually refers to "the number of misidentified fire

pixels in JMA-Himawari product, majorly caused by ground thermal sources", however, it only used "Number of fire pixels", which may make some misunderstanding. It will be better if Figure 6 (a) can have a clearer y axis.

**R: Thanks very much for this comment. This figure have been reproduced accordingly.**

**To Reviewer 2:**

The authors proposed an adaptive hourly NSMC-Himawari fire product for China based on the original Himawari data by employing a dynamical threshold for fire extraction and reference of ground thermal sources. Timely and accurate monitoring of wildfires is of great importance to inform management response and mitigation actions. This research and the derived dataset are a good contribution to the field. However, there are still some points that need to be addressed before publication.

**R: Thanks so much for all your constructive remarks and useful suggestions, which has significantly raised the quality of the manuscript. We have addressed the issues you raised in the response letter and the revised manuscript. By clarifying the issues you suggested, the manuscript has been largely improved. Thanks again for all your encouragement and valuable comments. Please feel free to contact us if additional revisions are required and we are more than willing to conduct further revisions according to your comments.**

**Major points:**

1. The authors' fire-identification approach for the NSMC-Himawari fire product is mainly based on their previous FY-3D global fire-identification algorithm. The major improvement is the dynamic adjustment of fire-identification thresholds. However, there remain some empirical parameters in Eq. (5). The sensitivity shall be tested and informed.

**R: Thanks so much for this comment. Actually, eq.(5) is the absolute condition to extract fire pixels. These three absolute conditions ($T_7 > 360$ K and $R_{vis} < 0.7$ and $\theta_{sz} > 87$ °) can be considered to be the borderline, such as 87 °, which is the zenith angle of the sun entering the twilight period. Once the observation period is in the twilight period, the accuracy will decline. 0.7 represents absolute high reflection, and 360 k is equivalent to 87 °C. When it is not in the twilight period, and there is no influence of high reflection surfaces and thermal factories, as long as it is not the observation noise, the pixels that meets these three conditions must be the fire pixels, because no other underlying surface can produce such strong energy. Therefore, based on a dataset of on-site collected burning records, we directly proved the effectiveness of our empirical parameters, instead of conducting sensitivity test here. In section 4.3, we detailed the verification results, and the final results showed that NSMC-Himawari-8 fire product was highly accurate in China, with the accuracy is 80 % and 84 % (not considering the omission errors), respectively, however, the accuracy of JMA-Himawari-8 fire product is only 54% and 59%, which is the direct evidence to prove the high accuracy of NSMC-Himawari-8 fire product and the effectiveness of our thresholds setting.**

**The clear explanation of absolute condition has been added to the revised manuscript. Thanks so much for this comment, which improved the rationality of this research significantly.**

2. Regarding the method verification, the authors proposed to evaluate the derived product using the JMA-Himawari fire products and on-site collected fire reference. Why not incorporate the FY-3D fire product as well? It will be a good benchmark to verify the algorithm improvement and product performance.

**R: Thanks so much for this comment. As stressed in the manuscript, NSMC-Himawari-8 fire product is based on the original observation data of Himawari-8 and adopts the more practical dynamic threshold algorithm for China and reprocesses fire pixels extracted, which is more suitable for China's vast territory and complex underlying surfaces, and has a high accuracy compared with JMA-Himawari-8 fire product. The two fire products are all based on the same satellite (Himawari-8) and have the same spatial resolution and temporal resolution, so consistency analysis can be carried out, and the accuracy comparison base on the dataset of on-site collected burning records is more convincing. Althouth the fire-identification approach for the NSMC-Himawari-8 fire product is based on our previous FY-3D global fire-identification, there are essential differences between them, as the spatiotemporal resolution and temporal resolution are completely different. Therefore, FY-3D is not incorporate.**

3. The explanation of some symbols is not clear enough. For example, in Table2, "$T_7$""$T_{13}$" and "$T_{14}$"need to indicate the specific bands, which can avoid ambiguity. In equation 10, the explanation for "$R_{vis}$" and "$R_{visbg}$" is same, so one of them must be wrong.

**R: Thanks so much for pointing this out. We acknowledged that we made a mistake on the interpretation of these parameters. $T_7$ is the brightness temperature at Band 7 (3.89 μm). $T_{13}$ is the brightness temperature at Band 13 (10.41 μm). $T_{14}$ is the brightness temperature at Band 14 (11.24 μm). $R_{vis}$ is the reflectivity of the high-temperature pixel at the visible-light band. $R_{visbg}$ is the averaged reflectivity of effective background pixels in the window at the visible-light band. We have corrected these problems accordingly in the revised.**

4. Some figures need to be improved. The legend in Figure 1 (b) is incomplete, and lacking of the fire area. The flowchart in Figure 2 is incomplete, lacking important contents in the "Data Reprocessing" process. In Figure 3, cloud coverage in the area of the nine-dotted line is only the cloud coverage of background, missing the actual cloud coverage. In Figure 6 (a), the title of the y-axis should be changed to "The number of misidentified fire pixels" to avoid confusion.

**R: Thanks so much for this comment. In the revised manuscript, we have corrected these figures accordingly.**

 **Minor points:**

1. Spell out the full name of JMA and NSMC in the abstract, since they appear for the first time.

**R: Thanks so much for pointing this out. In the revised manuscript, we added the complete name to promote readers' understanding and make the content of the article clearer.**

2. The time of the data (last access date) should be updated according to the requirement of ESSD.

**R: Corrected. Thanks so much for pointing this out.**

3. Please double-check the language throughout the manuscript. There are several typos and confusing parts. Some examples below:

Figure 1(a): "Hotintensity" ?

Figure 4 (a): "Mountian"

Line 229: "After removed the above mentioned disturbing pixels …"

**R: Thanks so much for this comment. We have corrected all these typos in the revised manuscript. Meanwhile, we have re-checked the manuscript carefully and polished the English. Thanks again for your comment.**

**List of all relevant changes made in the manuscript.**

| | |
|---|---|
| **Page 1** | 1. Revise some mistakes in grammar in abstract.
2. Update the time of the data (last access date).
3. Modified the file description (NSMC-Himawari-8 fire product).
4. Add the full names of NSMC and JMA.
5. Reduce irrelevant introduction and just retain some necessary and brief introduction of JMA-Himawari-8 fire product |
| **Page 3** | Reduced the introduction of GEOS-16 Advanced Baseline Imager (ABI) fire products, and just retained very limited information concerning this data set. |
| **Page 6** | 1. Revised the legend of Figure1.
2. Add the fire area in Figure1(b). |
| **Page 8** | Improve the flow chart, including all the significant processing steps (adding the removal of cloud pixels, the removal of the influence of cloud and desert edge and verification). |
| **Page 9** | 1. Supplement cloud coverage accuracy.
2. Correct the content of the nine-segment line in Figure 2. |
| **Page 10** | Correct the explanation for "$T_7$", "$T_{13}$" and "$T_{14}$" |
| **Page 11** | Revise some mistakes in "After we removed the above disturbing pixels in the window area …" |
| **Page 13** | Correct the explanation for "$R_{vis}$" and "$R_{visbg}$" in Eq(10). |
| **Page 15** | Correct the spelling error in Figure 4. |
| **Page 18** | Change the title of the y-axis in Figure6(a) to "The number of misidentified fire pixels". |